# IEKG: A Commonsense Knowledge Graph for Idiomatic Expressions

**Ziheng Zeng[1] Kellen Tan Cheng[2] Srihari Venkat Nanniyur[3] Jianing Zhou[1] Suma Bhat[1]**
[1]University of Illinois Urbana-Champaign [2]Princeton University
[3]Washington University in St. Louis
[1]{zzeng13,zjn1746,spbhat2}@illinois.edu
[2]kellentan@princeton.edu, [3]snanniyur@gmail.com

## Abstract

Idiomatic expression (IE) processing and comprehension have challenged pre-trained language models (PTLMs) because their meanings are non-compositional. Unlike prior works that enable IE comprehension through fine-tuning PTLMs with sentences containing IEs, in this work, we construct IEKG, a commonsense knowledge graph for figurative interpretations of IEs. This extends the established $\text{ATOMIC}^{20}_{20}$ (Hwang et al., 2021) graph, converting PTLMs into knowledge models (KMs) that encode and infer commonsense knowledge related to IE use. Experiments show that various PTLMs can be converted into KMs with IEKG. We verify the quality of IEKG and the ability of the trained KMs with automatic and human evaluation. Through applications in natural language understanding, we show that a PTLM injected with knowledge from IEKG exhibits improved IE comprehension ability and can generalize to IEs unseen during training.

## 1 Introduction

*Idiomatic expressions* (IEs) are frequently used in natural language and include a variety of versatile figures of speech that improve language fluency and conciseness in multiple genres (Moon et al., 1998; Baldwin and Kim, 2010; Haagsma et al., 2020). Prior NLP research in IE processing has focused on detecting idiomaticity ((Liu, 2019; Zeng and Bhat, 2021) among others), including tasks and applications that require IE comprehension, e.g., sentiment classification (Biddle et al., 2020), machine translation (Fadaee et al., 2018), natural language inference (Stowe et al., 2022; Chakrabarty et al., 2022b), and dialog systems (Jhamtani et al., 2021). IE comprehension poses challenges to NLP systems (Sag et al., 2002; Tayyar Madabushi et al., 2021; Stowe et al., 2022) mainly owing to their failure to account for IEs' characteristic *non-compositionality*, i.e., the meaning of an expression is not derivable from the meanings of its components (Baldwin and Kim, 2010). For example, consider the case of a sentiment classifier that incorrectly recognizes the negative sentiment in the statement *They have stirred up a hornet's nest*[1]. by failing to account for the figurative interpretation of the IE (Balahur et al., 2010). This study focuses on injecting IE-related knowledge into small-frame PTLMs known for their wide use, such as BERT (Devlin et al., 2019) and BART (Lewis et al., 2020), considering their struggle to understand the figurative meanings of IEs (Bhargava and Ng, 2022; Zeng and Bhat, 2022). We discuss the corresponding capabilities of large PTLMs, such as GPT-3.5, in the limitation section.

Towards enabling IE comprehension, prior efforts range from automatic idiom-detection methods (Liu and Hwa, 2019; Škvorc et al., 2022; Zhou et al., 2021) to learning IE representations, all using sentences with IEs (i.e., *idiomatic sentences*), where IE-specific information would be learned *implicitly* from limited samples and sparse contextual information (Zeng and Bhat, 2022). For instance, MAGPIE (Haagsma et al., 2020), the largest-to-date corpus for IEs, spans 1,755 distinct IEs, where a staggering 81% of them each have less than 50 idiomatic sentences (IEs are individually rare in natural language), and certain idiomatic sentences therein contain little to no contextual information to allow models to learn the IEs' meanings (e.g., *The thing was on the blink again.*). As such, prior work on IE embedding learning relied on the auxiliary aid of IE dictionary definitions to augment the information available in idiomatic sentences (Zeng and Bhat, 2022). An alternative approach of *explicitly* learning commonsense knowledge about IEs without significantly scaling the data/parameters constitutes the crux of this study. We rely on psycholinguistic findings about the impact of IE-related aspects, such as mental states, emotions, and likely

---

[1]The idiom "Stir up a hornet's nest" is used to convey the idea of causing trouble or inciting a commotion.

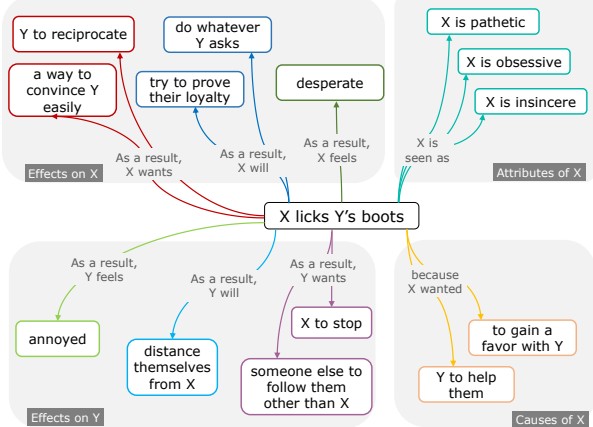

Figure 1: Example IE *lick someone's boot* in IEKG.

actions, on human IE comprehension (Rohani et al., 2012; Saban-Bezalel and Mashal, 2019), to explore the use of commonsense knowledge about IEs towards their comprehension.

Specifically, we build on the findings that commonsense *knowledge graphs* (KGs), e.g., ATOMIC$^{20}_{20}$ (Hwang et al., 2021), organized as if-then relations for inferential knowledge enable linguistic and social reasoning abilities for PTLMs (Bhargava and Ng, 2022). Indeed, models relying on their applications have benefited figurative language processing, such as their interpretation (Chakrabarty et al., 2022a) and generation (Chakrabarty et al., 2021b). However, the sparse IE coverage in these large-scale KGs limits their wider applicability for IE-related reasoning; as an illustration, ATOMIC$^{20}_{20}$ covers only 347 out of the 1,755 IEs ( 20%) in MAGPIE, with some contextually semantically ambiguous IEs only annotated with their literal sense while their figurative meanings are either missing or inaccurate. For example, for the IE *out of sight*, the only instance in ATOMIC$^{20}_{20}$ is <AtLocation, arctic>.

Hence, we construct the **I**diomatic **E**xpression **K**nowledge **G**raph (IEKG), an IE-centered extension to the ATOMIC$^{20}_{20}$ KG, that serves as an instance-efficient and explicit IE-related source of knowledge (compared to the implicit knowledge available in a large number of idiomatic sentences) for IE processing. This permits a novel exploration of how IE-related knowledge from a KG can be used for IE-related comprehension in parameter-efficient and sample-limited scenarios.

IEKG follows the schema of ATOMIC$^{20}_{20}$ (see Figure 1) and organizes the idiomatic interpretations for IEs into reasoning types covering the ef-

fects, causes, and attributes for both the subject and the object in an IE (see Section 3). Working closely with human annotators that created 56,315 instances for 1,229 idiomatic events and 11 relation types, IEKG's quality and diversity were then ascertained via human and automatic evaluation.

We exhibit its wide utility via (1) neural knowledge models (KMs) trained on IEKG to show IEKG equips PTLMs with IE knowledge that is generalizable to out-of-graph instances; and (2) IE knowledge injection to showcase the usefulness of IEKG for IE comprehension tasks, such as *natural language inference* (NLI) with IEs and *continuation classification* in which the model decides if a continuation is appropriate for a given context with an IE.

The contributions of this work are as follows.
(1) We propose IEKG, a commonsense knowledge graph focusing on idiomatically interpreted IEs;
(2) We show that IEKG transforms various PTLMs into IE-aware KMs to infer IE knowledge; compared to the ATOMIC$^{20}_{20}$ trained KM, IE-aware KMs can generalize better to unseen IEs (+22% METEOR (Lavie and Denkowski, 2009)) and to unseen relations for seen IEs (+30% METEOR) on IE knowledge tuple completion.
(3) We show IEKG endows PTLMs with improved IE comprehension ability; after injecting IEKG, PTLM achieves SOTA results on the IE NLI benchmark IMPLI (+12% accuracy), and exhibits a significant increase on the continuation classification task on the Figurative Narrative Benchmark compared to the baseline PTLM (+25%. accuracy)[2]

## 2 Related Work

**Datasets with Idiomatic Expressions.** Classic datasets for idiomatic sentences, such as VNC (Cook et al., 2008), SemEval5b (Korkontzelos et al., 2013), and MAGPIE (Haagsma et al., 2020), have been the primary source of implicit idiomatic knowledge in tasks such as IE identification (Zeng and Bhat, 2021; Škvorc et al., 2022), IE type disambiguation (Feldman and Peng, 2013; Rajani et al., 2014; Peng and Feldman, 2016; Salton et al., 2016; Liu and Hwa, 2017; Taslimipoor et al., 2018; Peng et al., 2014; Liu and Hwa, 2019), and IE representation learning (Zeng and Bhat, 2022). However, the contextual information available in these collections has been sparse and a

---

[2]IEKG data and the related code can be found at https://github.com/zzeng13/IEKG.

non-uniform number of instances per IE for LMs to learn their meanings from (Zeng and Bhat, 2022). Datasets of idiomatic sentences are available for specific tasks, such as machine translation (Fadaee et al., 2018), paraphrase generation (Zhou et al., 2021), natural language inference (Stowe et al., 2022; Chakrabarty et al., 2022b), and language generation (Chakrabarty et al., 2022a). This study presents an alternative general-purpose IE-related dataset as a knowledge graph, aiming to provide a more comprehensive, explicit, and instance-efficient way to represent IE usage.

**Commonsense Knowledge Graphs.** Knowledge graphs organize information into an ontology as a multi-relational graph of edges and entity nodes (Zou, 2020; Ji et al., 2022). Commonsense KGs, such as ATOMIC (Sap et al., 2019), ConceptNet (Speer et al., 2017), CSKG (Ilievski et al., 2021), ASER (Zhang et al., 2022), and TRANSOMCS (Zhang et al., 2020a), are a subset of KGs that focus on generic world commonsense knowledge. Specifically, ATOMIC (and its successor $ATOMIC^{20}_{20}$ (Hwang et al., 2021)) utilizes if-then relations to describe rich causal and inferential knowledge spanning several dimensions, including causes vs. effects, agents vs. themes, and voluntary vs. involuntary events. ATOMIC has also been used in figurative language-related applications, such as interpretation and generation of figurative language (e.g., simile, idioms, and metaphor) (Chakrabarty et al., 2021b, 2022a). However, as mentioned in Section 1, $ATOMIC^{20}_{20}$'s coverage of IEs is limited. Through the current work, IEKG extends $ATOMIC^{20}_{20}$'s representation to idiomatic events to further IE-related applications.

**Explicit Knowledge Injection using KGs.** Traditionally, entities in general KGs are converted into embeddings before being used for downstream applications (Wang et al., 2017; Dai et al., 2020). For commonsense KGs, most prior works follow the COMET paradigm (Bosselut et al., 2019), where the knowledge model (KM) is trained via the knowledge tuple completion task, where given a tuple `<head, relation, tail>`, the KM uses the head and relation to predict the tail. The KM, thus trained, becomes an on-demand commonsense knowledge query machine equipped with knowledge about any head entity. Additionally, it can generalize to previously unseen head entities or unseen relation types for a seen head entity, inferring additional information to aid tasks, e.g., language generation (Chakrabarty et al., 2021b, 2022a; Sabour et al., 2021). To inject explicit knowledge while preserving the PTLMs' ability as general-purpose LMs, prior works have continued LM pre-training on sentences converted from knowledge tuples (Chang et al., 2020; Agarwal et al., 2021). These broad schemes of KG-PTLMs combinations inspire our work to explore linguistic reasoning using the explicit knowledge in the IE-centric IEKG that we construct. Comparing various KG/PTLMs incorporation methods (Hu et al., 2022) constitutes a concrete direction for future work. Finally, others (Phang et al., 2018; Wang et al., 2020; Chang and Lu, 2021) use task-specific knowledge injection before or to replace fine-tuning when the task data is limited. We explore combining task-specific and IEKG knowledge injection (see Section 5) to enable IE comprehension on tasks with limited data.

# 3 Idiomatic Expression Knowledge Graph

By design, IEKG aligns with $ATOMIC^{20}_{20}$'s general structure and relation types to describe the linguistic knowledge of IE uses so that it enables commonsense inference on figurative language.

## 3.1 Expressions and Relation Types Selection

**Idiomatic event creation.** Given a collection of IEs, we first convert them into idiomatic events [3]. For example, the IE, *add fuel to the fire*, is converted to the idiomatic event, *PersonX adds fuel to the fire*. We use the 1,755 IEs from MAGPIE[4] as the initial set and apply a heuristic-based algorithm that automatically converts verbs and pronouns into appropriate and grammatically correct forms while adding subject/object placeholders, such as *PersonX*, to convert all the IEs to their corresponding idiomatic events. For the current scope of IEKG, we focus on IEs with persons as subjects and/or objects. Thus, in the second phase, two members of the research team went through the entire list of idiomatic events (1) to correct any grammatical errors in the idiomatic event and (2) to filter out IEs that cannot be used with persons as subject/object (conflicts were resolved via mutual agreements between the evaluators). Finally, 1,229 IEs (70%)

---

[3]"IE" and "idiomatic event" are equivalent hereafter due to the one-to-one mapping between them.

[4]Note that the IEs from MAGPIE are sourced from British National Corpus (BNC Consortium, 2007), and, as such, predominantly reflect their British usage.

are preserved in the IE collection with their corresponding idiomatic event created.

**Relation type selection.** To allow IEKG to become a natural extension to $ATOMIC_{20}^{20}$, we select a subset of relation types from $ATOMIC_{20}^{20}$ for IEKG. Several of $ATOMIC_{20}^{20}$ relation types are not suitable when the subject/object of a given event are persons, e.g., the relation type `is made of`. Hence, we exclude such relation types to select 11 relation types (See Table 5 in Appendix A) that are appropriate for the idiomatic events, covering, for instance, the intent, reaction, effect, causes, and attributes for the subject/object of the events, providing a multifaceted view of IE usage and interpretation (See Appendix D for a case study).

## 3.2 Knowledge Graph Construction

**Human annotators.** Prior efforts (Hwang et al., 2021; Haagsma et al., 2020) construct KGs and idiomatic sentence corpora via crowd-sourcing tools, such as Mechanical Turk, and control quality via numerous performance testing and ongoing monitoring schemes. For the construction of IEKG, three accountable, knowledgeable, and native English-speaking college students interested in an undergraduate research experience, volunteered their time. We communicated with the annotators throughout the annotation process, ensuring they understood the expectations, talking to them about research in NLP/ML with our research projects as examples, and, after our study, sharing our experimental results with them. Each annotator performed their annotation individually at their own pace with no time or monetization pressure to rush the annotation process. It may appear that our annotation would lack the diversity of typical crowd-sourced annotation; a diversity analysis (see Section 3.4) indicates that our resulting annotation was reasonably diverse. Thus, we believe that the gain in annotation quality in our non-crowd-sourced approach outweighs the loss in diversity.

**Annotation procedures.** To construct IEKG, for each idiomatic event, the annotators: (1) Confirm the understanding of the given IE and the validity of its idiomatic event by first looking up the IE and recording its dictionary definition and then verifying the idiomatic event is appropriate for the IE meaning and is grammatical. (2) Select the relation types, as applicable, to avoid forcing unreasonable annotations for unsuitable relation types. (3) Write annotations for each selected relation type; annota-

tors think of possible contexts and write up to *four* free-form phrases as tails, given the head (IE) and relations, to complete the annotations.

**Quality assurance.** We implemented various measures to ascertain annotation quality. During recruitment, we conducted annotator interviews to confirm their interest, background, and English proficiency as native speakers. Before the annotation phase, we organized info sessions on the research background and tutorials on the annotation procedure. We prepared detailed instructions with examples of completed annotations for ten events and four examples per relation type to demonstrate the definitions and differences among relation types. During the annotation phase, we held weekly office hours to provide clarification as needed.

## 3.3 IEKG Statistics

IEKG comprises 56,315 knowledge tuples, covering 1,229 idiomatic events and 11 relation types, with a mean annotation (tail) length of 2.98 words and a mean number of knowledge tuples per IE of 45.82 (standard deviation 13.67). On average, each IE has annotations for 7.41 applicable relation types (standard deviation 1.69).

Knowledge tuples are not evenly distributed over the 11 relation types. The most frequent relation type, `xEffect`, attributes to 18.63% of knowledge tuples, while the least frequent relation type, `oWant`, appears in only 2.63% of tuples. We account for this uneven distribution later in our sampling strategy for evaluations.

## 3.4 Annotation Quality Assessment

As shown in Table 1, IEKG provides more accurate idiomatic interpretations compared to $ATOMIC_{20}^{20}$. We perform a *human evaluation* and *diversity analysis* to rate the annotation quality.

**Human evaluation.** Members of the research team rated a sample of 500 knowledge tuples, with each sample evaluated by three members. To ensure the samples represent the overall quality, we mix annotations from all annotators and stratify sample tuples, preserving the original tuple distribution over the relation types. We use a 4-point Likert scale as our scoring metric (Hwang et al., 2021), reflecting how reasonable each knowledge tuple is and ranging from *Invalid* (score 4) to *Always/Often* (score 1), with a *lower* score indicating *higher* quality. We average the metric scores across human evaluators on each sample and then average across all samples. The final score is 1.51 (inter-annotator agreement

| Idiomatic Event | Relation | ATOMIC$_{20}^{20}$ | IEKG |
|---|---|---|---|
| PersonX buys the farm | xNeed | save up money | a tragic event |
| | xIntent | to grow crop and earn a living. | To pass away peacefully |
| | HinderedBy | The farm cost too much money. | a lack of immediate dangers |

Table 1: A comparison between ATOMIC$_{20}^{20}$ and IEKG for the *PersonX buys the farm* which means *die*. Unlike ATOMIC$_{20}^{20}$, IEKG provides accurate idiomatic interpretations.

is 50%), i.e., most annotations are either always or sometimes reasonable, indicating the annotations are of good quality.

**Diversity analysis.** Having established annotation quality, we show annotations have high *semantic* and *lexical* diversity across annotators as follows. First, for a given event and relation type annotated by all three annotators, we connect the tails from each annotator with "and" into three respective sentences. Then, with these triplets of annotation sentences, we compute the semantic diversity using average pairwise *BERT score* ([Zhang et al., 2020b](#)) and *embedding cosine similarity*[5] across all triplets of annotations. The resulting mean/std. BERT score after baseline scaling is 0.42/0.13, and the mean/std. embedding cosine similarity is 0.58/0.14, both indicating low semantic similarity among the annotations from different annotators. For lexical diversity, we concatenate each triplet of annotation sentences into a single sentence and compute the mean Google-BLEU score (i.e., mBLEU ([Zhang et al., 2020b](#))) that captures the distinctiveness of the n-grams across annotations and has a range between 0 (no match) and 1 (all matches). Our dataset shows an average score of 0.16, indicating a low lexical overlap across annotators.

## 4 Neural Knowledge Model for IEs

We experiment with KG/PTLM incorporation by training KMs on IEKG and comparing their capability in processing idiomatic events to the KM trained on ATOMIC$_{20}^{20}$ to showcase the additional IE knowledge that can be learned from IEKG.

### 4.1 Learning to Infer Knowledge

Our neural KM is a knowledge completion model that performs the *knowledge tuple completion task*. Various PTLMs that have a conditional generation ability (e.g., BART ([Lewis et al., 2020](#)), GPT-2 ([Radford et al., 2019](#)), and T5 ([Raffel et al., 2020](#))) and previously used for commonsense knowledge learning can be converted into KMs with common-

sense idiomatic knowledge after training on IEKG. We include the following baseline models (more details in Appendix [B](#)):

**BART-Comet** is the baseline 12-layer BART-large model trained on ATOMIC$_{20}^{20}$ and we use the trained checkpoint released by [Hwang et al. (2021)](#).

**BART/GPT2/T5-IEKG** are the 12-layer BART-large, GPT2, and T5 models fine-tuned on IEKG for the knowledge tuple completion task. We record the checkpoints with the best Rouge-L score on the test set during training.

### 4.2 Knowledge Model Experiments

**Data.** To evaluate a KM's ability to generalize to unseen relation types for seen idiomatic events and to new idiomatic events, we create two types of train/test splits of IEKG: (1) *relation-type split* and (2) *IE-type split*. For the relation-type split, we separate the relation types per idiomatic event and randomly assign a fraction of the relation types (and associated annotations) to a test set, retaining the rest in the train set, thereby ensuring that the two sets do not have overlapping relation types for the same events. This generalization ability is essential because IEKG's annotations for existing idiomatic events could be incomplete, as evidenced by the uneven distribution over the relation types (see [3.2](#)). For the IE-type split, we ensure no overlap between the IE types in the train and test sets and test a KMs' ability to generalize to IEs outside of the KG. This uses the explicit information from the KG and the implicit language knowledge encoded in the PTLM and accommodates the need to grow the KG to new IE types. We maintain an 80/20 train/test ratio: for the relation-type split, we have 45,103/11,212 training/testing tuples; for the IE-type split, we have 45,317/10,998 (983/246 IE types) training/testing tuples.

**Automatic Evaluation.** We used three widely used metrics for language generation to evaluate the tail generation quality, i.e., ROUGE ([Lin, 2004](#)), ME-TEOR, and BERTscore. For a given test event and relation, we took the top-1 KM-generated tail as the

---

[5]We used `sentence-transformer`'s best encoder (`all-mpnet-base-v2`) to generate sentence embedding.

| Model | BERT P | BERT R | BERT F1 | ROUGE-1 | ROUGE-L | METEOR |
|-------|--------|--------|---------|---------|---------|--------|
| BART-Comet | 0.8737 | 0.8742 | 0.8721 | 0.1576 | 0.1532 | 0.1189 |
| T5-IEKG | 0.9274 | 0.9342 | 0.9293 | 0.4415 | 0.4365 | 0.3488 |
| GPT2-IEKG | 0.9313 | 0.9285 | 0.9287 | 0.4420 | 0.4397 | 0.3069 |
| BART-IEKG | **0.9484** | **0.9465** | **0.9463** | **0.5688** | **0.5670** | **0.4161** |

(a) Performance on the relation-type split.

| Model | BERT P | BERT R | BERT F1 | ROUGE-1 | ROUGE-L | METEOR |
|-------|--------|--------|---------|---------|---------|--------|
| BART-Comet | 0.8745 | 0.8749 | 0.8729 | 0.1664 | 0.1640 | 0.1262 |
| T5-IEKG | 0.9202 | 0.9277 | 0.9222 | 0.3800 | 0.3761 | 0.2914 |
| GPT2-IEKG | 0.9254 | 0.9218 | 0.9222 | 0.3902 | 0.3893 | 0.2711 |
| BART-IEKG | **0.9401** | **0.9377** | **0.9377** | **0.4937** | **0.4905** | **0.3474** |

(b) Performance on the IE-type split.

Table 2: Knowledge models performances by automatic metrics, including BERT score precision (BERT P), recall (BERT R), F1 (BERT F1), ROUGE-1, ROUGE-L, and METEOR score. Best performances are **boldfaced**.

candidate and used all corresponding annotations as references to compute the scores.

**Human Evaluation.** Three annotators of IEKG manually evaluated the tail generation quality. We utilize the same 4-point Likert scoring metric for annotation quality assessment as detailed in Section 3.4. Finally, we average the manual scores per sample and then average over the instances. We sample 100 generated instances from the relation-type test split of BART-Comet (indifferent to IE/relation-type split) and BART-IEKG; and 100 generated instances from the IE-type test split of BART-IEKG. Comparing BART-Comet's generated instances against BART-IEKG's, we assess the quality gain from IEKG. Evaluating BART-IEKG's generated instances on the IE-type split, we assess its ability to generalize to unseen IEs.

### 4.3 Results and Discussion

As shown in Table 2, and unsurprisingly, all PTLMs trained on IEKG substantially outperform the baseline BART-Comet in the tuple completion task across all metrics and both splits. Compared to BART-Comet, BART-IEKG attains absolute gains of 7.4% in BERT F1, 41% (260% relative) in ROUGE-L, and 27% (225% relative) in METEOR score in *relation-type* split. Considering an IE-type split, it achieves a comparable gain over BART-Comet with a gain of 6.5% in BERT F1, 33% (201% relative) in ROUGE-L, and 22% (174%) in METEOR score. Besides, BART-IEKG outperforms GPT-2 and T5-based models (which is consistent with the results from Hwang et al. (2021)) achieving the best performances across both splits.

Then, we compare the best-performing BART-IEKG model and the baseline BART-Comet via human evaluation. Quantitatively, for the test sam-

ples from the relation-type split, the average human evaluation score for BART-Comet is 2.12, while the score for BART-IEKG is 1.65, gaining around 0.47 points (lower is better). To put this gain into perspective, we also check the human evaluators' preferences between the outputs from BART-Comet and BART-IEKG. Specifically, given a pair of outputs from the same test instance, we say BART-IEKG's output is preferred over BART-Comet's output if the BART-IEKG's score is strictly smaller or equals 1 (the best score) by two or more annotators out of three. The evaluators prefer BART-IEKG's outputs on 74% of the test samples. The average evaluation score for BART-IEKG's outputs on the IE-type split is 1.64, suggesting that BART-IEKG can generalize to unseen IEs and produce reasonable outputs (based on the mean score is less than 2) according to human judgment. See Appendix C for more example generations.

## 5 Applications

We show how IEKG enables IE comprehension through two applications that require a model to understand IEs' semantics to perform correctly with their largest and most recent benchmark datasets.

### 5.1 IEKG Injection

Many prior works (e.g., (Chakrabarty et al., 2021a)) combine input sentences and additional commonsense information queried from a KM to perform downstream tasks. However, this incurs significant computational overhead and does not endow the PTLM with an innate IE comprehension ability. Instead, we fine-tune PTLMs using the mask-infilling objective to imbue a pretrained BART model with

explicit IE-related knowledge from the IEKG[6], similar to Agarwal et al. (2021). First, we transform each knowledge tuple into a two-mask template. For example, the knowledge tuple <*PersonX wins by a hair's breadth, xAttr, Ambitious*> is converted into *In PersonX wins by a hair's breadth, a hair's breadth means* <MASK>. *PersonX is seen as* <MASK>. The model must infill the first mask with the IE definition and the second mask with the appropriate inference given the relation type (*xAttr* in this example). Unlike KMs, IEKG injection allows the PTLM to learn the IE semantics and related commonsense knowledge while preserving its ability to be further fine-tuned for other tasks.

## 5.2 Tasks in the presence of IEs

**Natural Language Inference (NLI)** involves determining if a given hypothesis agrees with a given premise (entailment, denoted as E) or not (non-entailment, denoted as NE). Prior work (Stowe et al., 2022) shows that the presence of IEs degrades NLI performance. Thus, our first objective is to examine whether IEKG injection benefits NLI in the presence of IEs. Specifically, we use the Idiomatic and Metaphoric Paired Language Inference (IMPLI) (Stowe et al., 2022) dataset. For training, we use IMPLI's *silver* split, where samples are taken from MAGPIE, PIE, and SemEval, with labels created automatically, to harness their relative abundance compared to the test set (see below). We randomly select an equal number of entailment and non-entailment samples to balance the classes. For testing, we use IMPLI's *gold* split; these are manually generated samples and are hence more reflective of true model performance.

As a second objective, we study the individual effects and interactions of task-specific knowledge injection and task-specific fine-tuning of PTLMs. This process quantifies how much of the performance gain, if any, can be explained by only task-specific knowledge injection, and how much can be attributed to our IEKG injection. Specifically, some baselines may also be fine-tuned on the MNLI dataset (Williams et al., 2018), with balanced entailment and non-entailment samples. Overall, we consider 5 possible settings: vanilla PTLM (BART-large), PTLM fine-tuned on MNLI, PTLM with IEKG injection, PTLM fine-tuned on MNLI followed by IEKG injection, and PTLM with IEKG

injection followed by fine-tuning on MNLI.

**Context continuation** involves selecting the continuation sentence that makes the most sense with the provided context with an IE. Prior work (Chakrabarty et al., 2021a) illustrated this task's difficulty even for very large models, e.g., GPT-3. We aim to determine whether IEKG injection helps identify correct continuations. Specifically, we use the Figurative Narrative Benchmark (FNB) dataset (Chakrabarty et al., 2021a), where continuation selection requires correct IE interpretation. Each instance consists of a given context with an IE, followed by one correct and one incorrect continuation. We transformed this dataset into a binary classification task by generating two examples (an incorrect and correct continuation) from each instance in the dataset, formulated as <context, continuation>. The model must then classify whether the example is correct. We analyze the performance of the idiomatic samples' test split, where the labels are graciously provided by the authors of FNB (Chakrabarty et al., 2021a).

Note that in the above tasks, we have included the two largest and most well-defined publicly available datasets for figurative language comprehension at the moment. In Appendix H, we also describe experiments conducted on two additional datasets for IE comprehension. In these experiments, we observe results similar to those presented in the main paper: models equipped with IEKG knowledge injection exhibit enhanced figurative language comprehension abilities. We did not include generic NLU datasets like SNLI, MNLI, or SST because these datasets are not specifically designed to assess NLU in the presence of idiomatic expressions.

## 5.3 Experimental Setup

Fine-tuning uses a pretrained BART-large model, with all experiments utilizing the same hyperparameter settings to maintain consistency across our findings. Further details can be found in Appendix E.

## 5.4 Application Results

Our first objective is to quantify the benefits, if any, of IEKG injection on both the NLI and continuation tasks in the presence of IEs. On the NLI task, Table 3 illustrates that IEKG injection results in significant gains in performance. BART-IEKG outperforms the pretrained BART by 2.33%, while BART-IEKG-MNLI achieves significant gains of 18.01%. Most importantly, combining IEKG injection with

---

[6]We also found IEKG-trained KMs are useful for downstream IE comprehension tasks. See Appendix H.

task-specific fine-tuning, which is used to help the model understand the NLI task itself, results in state-of-the-art performance. BART-MNLI-IEKG scores 83.75% on the IMPLI gold samples, which represents a 23.68% increase in accuracy compared to a pretrained BART mode and is still 5.79% better than BART-MNLI. These performance gains may be partly explained if the MNLI dataset is similar to the IMPLI gold samples, which are out-of-distribution (OOD) to the silver samples (as the vanilla BART-large model performs poorly during testing). Fine-tuning on external datasets similar to the OOD dataset has been shown to increase model robustness (Liu et al., 2022). However, MNLI alone cannot fully explain our overall performance, as IEKG injection brings additional improvements. As the IEKG step utilizes the mask-infilling objective, it cannot be the case that this step is similar to how we perform inference on the IMPLI gold samples (the IEKG step is a generation, not classification). Thus, we conclude that task-specific fine-tuning by itself is inadequate while additional IE knowledge introduced by IEKG injection is necessary to achieve the best performance.

Notably, our best-performing model, BART-MNLI-IEKG, achieves a *non-entailment* accuracy of 68.11%, which is almost double the previous SOTA (34.80%) published with IMPLI (Stowe et al., 2022), outperforming both BART (+15.35%) and BART-MNLI (+33.07%) in accuracy.

Non-entailment performance is a crucial barometer of true comprehension, as entailment is often easily classified for all models (this may be explained by high token overlap or predicting entailment as the majority class). That is why baseline models perform well on entailment samples but struggle greatly on the non-entailment and antonym non-entailment samples (see Appendix F for examples). While poor non-entailment performance was identified as a major challenge by the authors of the IMPLI dataset (Stowe et al., 2022), our results demonstrate that IEKG can significantly mitigate this challenge. BART-MNLI-IEKG exhibits strong performance across the board, and its high non-entailment performance implies that our model is truly comprehending the samples instead of applying rudimentary heuristics such as token overlap.

On the FNB test split, Table 4 shows that IEKG injection results in a gain of 25.16% in overall performance compared to the baseline. Note that while state-of-the-art is around 83%, its method is much more involved, as it includes both generation and classification, and is followed by what is essentially a 12 inference ensemble for each sample (Chakrabarty et al., 2021a). Our results still demonstrate IEKG utility, as BART-IEKG can perform quite well without making unnecessary architecture adjustments or creating large ensembles. Additionally, across both datasets, IEKG injection has shown to be widely applicable to any tasks with IEs, and only requires a single fine-tuning step to imbue models with this knowledge.

Interestingly, IEKG injection provides a performance gain across IEs covered *and* not covered by IEKG. There are marginal gains in the performance of IEKG-covered samples on IMPLI (84.65% v. 82.39%) and comparable performance in accuracy on the FNB dataset (77.77% v. 77.81%). The performance on the uncovered idioms was particularly impressive compared to models without IEKG injection. Examining this phenomenon more closely, we had BART-IEKG generate definitions for a large set of uncovered IEs to understand how well the model could comprehend unseen IEs. Additionally, we wanted to see if the unseen IEs were very similar to the IEs present in IEKG. Similarity for these unseen idioms was computed in *token-wise* and *semantic-wise* similarity with idioms from IEKG. We found little correlation, if any, between the definition quality (as evaluated using the BERTScore) and either token-wise or semantic-wise similarity. Uncovered idioms similar to covered idioms exhibited higher quality definitions, but the converse was not necessarily true. See Appendix G for a detailed analysis.

The trends described in this section also hold for the BART-base models (See Appendix F).

## 6 Conclusion

We propose IEKG, a knowledge graph that describes the commonsensical, figurative interpretation of IEs. We show various PTLMs can be converted into KMs via the tuple completion task by training on IEKG and IEKG imparts useful utility for tasks that require IE comprehension.

Future work should extend IEKG to disambiguate between literal and figurative IE interpretations, and investigate factors that affect the models' generalizability to IEs not covered by IEKG.

| Model | Overall Acc. | E Acc. | NE Acc. | ANT Acc. |
|---|---|---|---|---|
| BART | 60.07% | 95.83% | 52.76% | 14.67% |
| BART-IEKG | 62.40% | 96.97% | 48.82% | 22.93% |
| BART-MNLI | 77.96% | 96.21% | 35.04% | **81.33**% |
| BART-MNLI-IEKG | **83.75**% | 96.02% | **68.11**% | 77.07% |
| BART-IEKG-MNLI | 78.08% | **97.53**% | 32.81% | 81.28% |

Table 3: A comparison of different BART-large models on the IMPLI dataset. E denotes entailment, NE denotes non-entailment, and ANT denotes antonym non-entailment samples. The best performing scores are **bolded**.

| Model | Acc. | CC Acc. | IC Acc. |
|---|---|---|---|
| BART | 52.63% | 52.40% | 52.92% |
| BART-IEKG | **77.79**% | **78.47**% | **77.11**% |

Table 4: A comparison of the effect of IEKG injection with a BART-large model on the FigurativeNarrativeBenchmark test dataset. Performances for correct (CC) and incorrect (IC) classes are shown. The best performing scores are **bolded**.

## Limitations

The main limitation of the current IEKG is that it only provides figurative interpretations of IEs. However, IEs are often contextually ambiguous, i.e., certain IEs can be interpreted literally in specific contexts. Hence, one future direction would be to find ways to include literal IE interpretations and distinguishing them from the figurative semantics. Second, this work focuses on enabling smaller frame PTLMs (with only millions of parameters) targeting the specific ability of IE comprehension using a data-efficient method. We are aware of and have experimented with larger PTLMs, e.g., GPT-3.5 and GPT-4, and noted their IE comprehension ability. For example, given appropriate prompts, GPT-3.5 can retrieve the definition for a given idiomatic event and generate reasonable, commonsense knowledge for a given relation type. However, given the extensive data and computing resources required by models like GPT-3.5, we believe that studying the ability of larger PTLMs is beyond the scope of the present work. Instead, GPT-3.5 could be used to collect and annotate efficiently IEKG-like knowledge graphs for more IEs, reducing the workload and the requirement of human annotators. Due to its unavailability during the time of data annotation for IEKG, we did not use GPT-3.5 to produce an even larger KG.

Additionally, our paper used the NLI and continuation classification tasks to understand the effect of IEKG injection on IE comprehension. It may be relevant for future endeavors to explore other tasks such as natural language generation. It would also be interesting to know whether larger models could also exhibit performance gains on OOD idioms, as we saw in our experiments. Specifically, it would be useful to examine if larger models could better explain this increased robustness that occurs with IEKG injection. However, due to their inherent complexity and costs, we did not make use of these larger models for testing IEKG injection in our experiments.

Finally, hyperparameter testing was not optimized for each individual step in a model. It is our belief that while optimizing hyperparameters should yield higher performance, the trends between different pipelines should remain the same (i.e., IEKG injection should still be beneficial). Nonetheless, it may also be possible in any experimental setting to achieve better performance with better-tuned hyperparameters.

## Ethics Statement

In this work, we proposed and constructed a knowledge graph dataset, IEKG. During data collection (Section 3, trustworthy undergraduate annotators interested in participating in a research experience in the field of NLP, were fully aware of the motivation and intent of the dataset and volunteered their time in creating the dataset while also learning about SOTA NLP models and their challenges. As the annotators are the sole creators of the dataset, there is no concern about infringing existing intellectual property or violating privacy rights. The annotators were instructed not to create explicit or toxic data. During our evaluation of the annotation and case studies, we found no data instances that contain toxic language. To further ensure data integrity, we combine annotations for the same event and relation type into single sentences and used a hate speech detector (Vidgen et al., 2021) that covers various types of hate to evaluate the toxicity. As a result, only 0.066% (103 out of 15,684 combined annotation sentences) of

the annotations contain potential hate speech. Manual inspection reveals that these samples contain annotations for idioms that describe PersonX negatively in nature, such as being "ignorant" and "foolish," and no actual targeted hate speech or toxic expressions were found. Hence, we believe that our dataset is safe for release. Additionally, the annotators are native speakers of standard American English and would not have accounted for differences in background/common knowledge of idiom-related attributes from other English-speaking cultures (Acharya et al., 2021).

For our applications in IE comprehension tasks, the intended use case is for the semantic analysis in the presence of IEs, which traditionally would cause misunderstanding for smaller-frame language models. In our experiments, only peer-reviewed and publicly available datasets are used, and no data contains sensitive personal/private information that could potentially breach privacy rights. We did not study biases in our experiments, but we do not anticipate there to be additional biases outside of those that pervasively exist in all facets of the English language (our experiments only focused on English datasets). The failure of our model would cause misinterpretation of IE's figurative semantics and thus may result in incorrect classifications in IE-related natural language understanding tasks. We do not anticipate these models to be deployed in high-risk environments (e.g., financial, medical and culturally-sensitive settings) as yet, but if they are, users should beware that model failure could cause semantic misinterpretation which could be detrimental depending on the actual use case. Finally, we ran moderately sized models that are orders of magnitude smaller than models such as GPT-3.5 or GPT-4. Thus, our environmental impact throughout the process is minimal, and run times were not overly excessive.

## Acknowledgements

We acknowledge and thank the assistance of Razi Ahmed Khan, Andrew Zhang, and Zachary Kuo in creating and curating the knowledge graph. This research was supported in part by the National Science Foundation under Grant No. IIS 2230817 and by a U.S. National Science Foundation and Institute of Education Sciences grant (2229612).

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

| Relation Type | Meaning |
|---|---|
| Causes | causes |
| HinderedBy | Can be hindered by |
| oEffect | As a result, Y or others will |
| oReact | As a result, Y or others feels |
| oWant | As a result, Y or others want |
| xAttr | X is seen as |
| xEffect | As a result, PersonX will |
| xIntent | Because PersonX wanted |
| xNeed | But before, PersonX needed |
| xReact | As a result, PersonX feels |
| xWant | As a result, PersonX wants |

Table 5: Relation types in IEKG and meanings.

## A    Relation Types in IEKG

IEKG is constructed to align with the $\text{ATOMIC}^{20}_{20}$ such that the relation types in IEKG is a subset of that in $\text{ATOMIC}^{20}_{20}$. Please refer to Table 5 for the 11 relation types in IEKG and their meanings.

## B    Knowledge Model Experiment Details

In this section, we provide details on the baseline knowledge models and experiment settings discussed in Section 4.

**BART-Comet** is the pre-trained BART-large language model with 12 Transformer encoder-decoder layers trained on the $\text{ATOMIC}^{20}_{20}$ KG. BART's encoder-decoder structure and denoising pre-training objective make it suitable for language generation tasks and has been shown to achieve the best performance among the KG-adapted PTLMs. We use the trained checkpoint released by Hwang et al. (2021). This model serves as a baseline for idiomatic knowledge inference.

**BART-IEKG** is a pre-trained BART-large language model that is fine-tuned on IEKG using the knowledge tuple completion.

**GPT2-IEKG** is an auto-regressive language model with 12 Transformer decoder layers, fine-tuned into a KM using IEKG with knowledge tuple completion task.

**T5-IEKG** is another 12-layer encoder-decoder transformer-based LM that is pre-trained with multiple NLP tasks; we fine-tuned the T5 model similar to BART-IEKG.

**Experiment Setup.** For all the PTLMs in our experiments, we use the HuggingFace implementations (Wolf et al., 2020). We train the models (except for BART-Comet, which is already fine-tuned) on IEKG for 50 epochs with a batch size of 16

for models with BART as the backbone and 64 for other models. For all training, we used the AdamW optimizer with a learning rate of 1e-5. All the other hyper-parameters are set to their default values. We record the checkpoints with the best Rouge-L score on the test set during training.

## C    Examples of Generated tuples from BART-Comet and BART-IEKG

In this section, we show a sample of generated knowledge tuples from BART-Comet and BART-IEKG for the relation-type split (Table 6) and IE-type split (Table 7). As shown in Table 6, observing example outputs from the relation-type test split, we find that BART-Comet often considers IEs literally when the figurative meaning is not understood while BART-IEKG generates correct figurative interpretations for the idiomatic events. Moreover, BART-IEKG can generalize to certain IEs unseen during training, as shown by the first three rows in Table 7; though when the BART-IEKG fails to understand the IE semantic, it tends to also produce literal interpretations as shown by the last three rows in Table 7.

## D    IE Grouping with IEKG

Grouping semantically related expressions is frequently encountered in NLP and naturally applicable to IEs, but remains under-explored owing to the challenges in processing their non-compositionality. Recently, Zeng and Bhat (2022) have used IE definition embeddings to group IEs based on their semantic relatedness. We propose an alternative harnessing the IE dimensions conveyed by the relations in IEKG, useful for downstream applications such as paraphrasing, e.g., by finding IEs with similar intents (the xIntent relation).

**Grouping methods.** We use *Definition Grouping* as a baseline, which computes the definition embedding for each IE using a sentence embedding model[7]. Then, for each IE, we find the closest related IEs from the lexicon by computing the pairwise cosine similarity of all IE definition embedding pairs and identifying the IEs with the highest cosine similarities. For *IEKG Grouping*, we identify related IEs by each relation type. First, we generate embeddings for all knowledge tuples in IEKG. To better utilize the sentence embedding model, we convert a given knowledge tuple into a

---

[7]We use the same sentence encoder from our diversity analysis.

| Idiomatic Event | Relation | BART-Comet | BART-IEKG |
|---|---|---|---|
| PersonX tells tales out of school | Causes | PersonX is not allowed to tell tales at school. | others to dislike them |
| PersonX gets down to brass tacks | HinderedBy | PersonX doesn't have the time to get down to brass tacks. | distractions |
| PersonX turns a blind eye | oReact | angry at personx for ignoring them. | grateful |
| PersonX is out of the running | xNeed | to not want to be in the running | to be in a competition |
| PersonX picks holes | HinderedBy | PersonX doesn't know how to pick holes. | having a strong work ethic |
| PersonX gets their feet wet | xIntent | to play in the water | to get better at something |
| PersonX is treading water | xNeed | to get in the water | a lack of interest |
| PersonX sails close to the wind | oReact | like they have a good friend to sail with. | scared |
| PersonX rolls with the punches | xIntent | to be a good person | to prove themselves |
| PersonX is on the level | xIntent | to be on the same level as everyone else | to be fair |

Table 6: Knowledge tuples completed by BART-Comet and BART-IEKG on relation-type split.

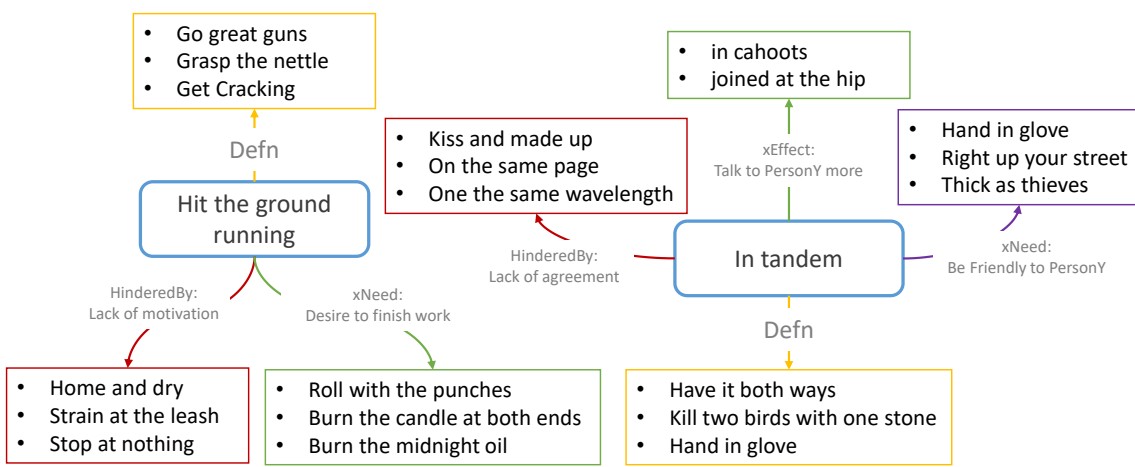

Figure 2: IEs grouped by their definition embeddings (Defn) and relation types from IEKG. IEs from IEKG grouping may not be semantically interchangeable but share similar attributes by relations.

| Idiomatic Event | Relation | Output |
|---|---|---|
| PersonX drops the ball | xAttr | unreliable |
| PersonX reaches for the stars | HinderedBy | laziness |
| PersonX is on thin ice | xWant | someone to help them |
| PersonX rub shoulders with PersonY | xAttr | strange |
| PersonX falls asleep at the wheel | xReact | sleepy |
| PersonX is tilting at windmills | xEffect | consult experts |

Table 7: Knowledge tuples completed by BART-IEKG on the IE-type splits. The first three rows show cases where the KM generalizes to unseen IEs while the last three rows show cases where the IEs are not correctly comprehended.

natural sentence by replacing its relation type with its corresponding text as listed in Table 5 and then concatenate the idiomatic event, translated relation type, and its tail into a sentence; then, we use the sentence embedding model to generate an embed-

ding for the tuple. Next, for any two given IEs from IEKG, $I_1$ and $I_2$, and each relation type $r$, we gather their sets of knowledge tuple embeddings for $r$ as $\mathbf{T}_{I_1}^r$ and $\mathbf{T}_{I_2}^r$; then, we compute the mean pairwise cosine similarity for pairs from $\mathbf{T}_{I_1}^r$ and $\mathbf{T}_{I_2}^r$ as the similarity measure for $I_1$ and $I_2$ w.r.t. $r$. In the end, for any given IE, we can find a list of the closest related IEs for each relation type in IEKG.

**Case study.** In Figure 2, we show the most closely related IEs identified via Definition Grouping and IEKG Grouping for various relation types. For example, for the IE *in tandem*, the most similar IEs found via Definition Grouping are *have it both ways* and *hand in glove*, which are mostly inappropriate. From IEKG Grouping, however, *kiss and make up* and *on the same page* are identified via the relation HinderedBy because *a lack of mutual goals or agreements* or *dislike of each other* will hinder these IEs; *in cahoots* and *joined at the hip* are identified via the relation xNeed because the

subject needs to *have a fondness* for the object in all three IEs. These results show that IEKG enables the IEs to be related to each other beyond semantics from their definitions but also by the shared attributes along each relation type, thus allowing for a more comprehensive IE understanding.

## E  Settings for Application Experiments

All fine-tuning steps are conducted at $5$ epochs, with a learning rate of $2e - 5$, and a weight decay of $0.01$. The batch size is $32$. All experiments use a random seed of $42$.

For the IEKG injection step, we use beam-search with $4$ beams, and a maximum new token count of $256$, and a minimum new token count of $10$. The knowledge tuple templates themselves are set at a maximum token length of $128$, with padding set to the maximum length, and truncation enabled. The dataset consists of $26,020$ templates in the fine-tuning step. The samples are organised such that the model sees all possible idioms in the IEKG during training, but we withhold some of the relation types for each idiom. For example, if a particular idiom had $7$ possible relation types, the model may only see $2$ or $3$ of these relation types during the IEKG injection step. Note that different relation types are withheld for each idiom to ensure that all relation types are seen across the entirety of the fine-tuning dataset. This ensures that the model both learns the idiom definition and how to apply the idiom semantics towards IE comprehension.

The MNLI fine-tuning step uses $258,514$ samples to imbue the model with task-level knowledge about NLI. These samples are split evenly between $129,257$ samples each for entailment and non-entailment, to ensure against classifier variability due to class imbalance. The samples are set at a maximum token length of $256$, with padding set to the maximum length, and truncation enabled.

The IMPLI step uses $13,650$ samples as the silver fine-tuning dataset. There are $6,825$ entailment samples, and $6,825$ non-entailment plus antonym non-entailment samples. In the IMPLI gold dataset, there are a total of $1,157$ samples, with $528$ being entailment, $254$ being non-entailment, and $375$ being antonym non-entailment samples. On a quick note regarding coverage, $697$ of the gold samples contain IEs that are present in the IEKG knowledge base. Of these $697$ samples, $317$ are entailment, $147$ are non-entailment, and $233$ are antonym non-entailment. The coverage of IEKG on the IMPLI gold is thus $60.24\%$.

The FNB step uses $6,408$ examples as the fine-tuning dataset, taken from an original amount of $3,204$ samples in the FNB training split. The FNB test split consists of $3,084$ examples, taken from an original amount of $1,542$ samples in the FNB test split. Regarding coverage, $902$ of the $1,542$ test samples contain IEs that are present in the IEKG knowledge base. The coverage of IEKG on the FNB test split is thus $58.50\%$.

## F  BART-Base Model for IMPLI and FNB

We observe that on the IMPLI gold dataset, the BART model with IEKG injection preceded by task-level MNLI fine-tuning still outperforms its contemporaries, with a gain of $2.15\%$ compared to the BART model that was only fine-tuned on MNLI, as evidenced by Table 8. There is still a significant gain compared to models without IEKG injection and MNLI fine-tuning, as it outperforms a vanilla BART model by $15.12\%$. Notably, even in the BART-base scenario, we still demonstrate a significant gain on the non-entailment performance, doing approximately $25\%$ better than the original IM-PLI state-of-the-art, and outperforms its other contemporaries by at least $7.48\%$ here. This demonstrates that despite model size difference, IEKG injection represents a significant advancement in model comprehension towards understanding the non-entailment examples that hitherto state-of-the-art models struggled on. These trends are exactly what we saw with our results on the BART-large models in Section 5.4.

Additionally, Table 8 shows that the vanilla pre-trained BART-base model also struggles on the non-entailment and antonym non-entailment samples. As mentioned previously in Section 5.4, entailment samples are easy to classify, as the pre-trained BART model sees the high token overlap and will predominantly predict entailment. To give one such example, consider the IMPLI entailment sample with the premise and hypothesis as follows: $<$*The Book of Proverbs makes it clear that happiness and discipline go hand in hand from the beginning of our lives : 'He who spares the rod hates his son, but he who loves him is careful to discipline him.', The Book of Proverbs makes it clear that happiness and discipline are associated with the beginning of our lives : 'He who spares the rod hates his son, but he who loves him is careful to discipline him.'$>$. With so many overlapping

tokens, it is easy for the pretrained BART model to employ a rudimentary heuristic like token overlap to predict entailment. However, this strategy fails on the non-entailment and antonym non-entailment samples (antonym non-entailment specifically uses the opposite meaning of the IE as opposed to just a random meaning), where the premise and hypothesis may still have high token overlap, but correct classification is now dependent on true model comprehension of the IE. For the non-entailment sample: <*It was pretty interesting, like one of the guys goes native and one of the guys doesn't.*, *It was pretty interesting, like one of the guys goes wild and one of the guys doesn't.*>, there is still high token overlap, but the two sentences are actually not in entailment. However, as the pretrained BART model still predicts entailment, the model fails at true IE comprehension. Similarly, this is also the case for the antonym non-entailment samples, with one such sample as follows: <*He prefers acting with other countries to going it alone.*, *He prefers acting with other countries to doing it with all the assistance of others.*>. The model's overall poor performance on non-entailment and antonym non-entailment samples, at both the BART-base and BART-large cases, indicates that pretrained BART models are not actually comprehending the IE meaning in the samples, which is why our IEKG injection step is so necessary.

Similarly to the BART-large case, we see that for the FNB test split, the IEKG injected model still outperforms the vanilla BART model, with a performance gain of 1.66% as seen in Table 9. It is interesting to note that the BART-base models here tend to perform better than the BART-large models, at least in the vanilla case. We hypothesize that at this model size, the models aren't able to truly separate since the size constrains the model's ability for IE comprehension, and is why at larger sizes we see a much more significant gap. Regardless, we see that IEKG injected models still outperform their counterparts across the board on the FNB test split, which follows the trends we saw in Section 5.4.

Finally, we see that, similarly to our results on the BART-large models, that IEKG injection seems to benefit both samples covered and not covered by IEKG, which adds a level of robustness and makes the model more generaliseable. Once again, we suspect that at such small model complexities, separability between models with and without IEKG is more difficult, as the model size itself is a constrain-

ing factor in IE comprehension. Our results still show a significant gap on the BART-large models however, and still demonstrate some separability, at least for the IMPLI results.

## G Analysis of IEKG Injection Robustness

In both the BART-base and BART-large cases, Table 11 and Table 10 indicate that performance on uncovered idiom samples appears to be pretty good. To examine why IEKG injection increases model generalizability, we tested the definition generation quality of our IEKG-injected BART-large model. We used a dataset of 2,166 idioms, 937 of which are not found in the IEKG knowledge base. As ground truth definitions are automatically pulled from the web (Wikitionary and Google Dictionary), they inherently carry some noise. After removing invalid ground truth definitions (e.g., definitions of the format "alternative spelling of"/"alternate form of"), the result is a total of 899 valid, uncovered idioms.

We then define two notions of similarity that can be used to compare uncovered versus covered idioms. First, token-wise similarity computes the idiom embeddings by passing the idiom tokens through a sentence embedding model (all-mpnet-base-v2 from sentence-transformer package) model, which generates a $(2166, 768)$ idiom embedding matrix. We then compute the pairwise similarity matrix, of shape $(2166, 2166)$, using the cosine distance as the metric of similarity. From here, each idiom is assigned a similarity score, computed as the mean of the top-3 highest values for that idiom's entry in the pairwise similarity matrix. For token-wise similarity, we aim to test if the model better generalizes to the unseen idioms that share token-level similarity with the seen idioms. Note that scores are bounded between $0$ and $1$, with larger scores indicating higher similarity and vice versa. The same procedure is followed in semantic-wise similarity, except that the idiom embedding is now computed from the idiom's ground truth definition, as opposed to the tokens. For semantic-wise similarity, we test if the model generalizes to unseen idioms with seen figurative semantics better.

As we explained briefly in Section 5.4, there is little correlation between generated definition quality and either token-wise similarity or semantic-wise similarity. From Table 12, we see a marginal correlation between the definition quality and

| Model | Overall Acc. | E Acc. | NE Acc. | ANT Acc. |
|---|---|---|---|---|
| BART | 56.27% | 89.20% | 51.97% | 12.80% |
| BART-IEKG | 56.87% | **93.37%** | 48.82% | 10.93% |
| BART-MNLI | 69.24% | 86.91% | 30.83% | 70.32% |
| BART-MNLI-IEKG | **71.39%** | 92.05% | **59.45%** | 50.40% |
| BART-IEKG-MNLI | 70.80% | 88.61% | 33.99% | **70.59%** |

Table 8: A comparison of different BART-base models on the IMPLI dataset. E denotes entailment, NE denotes non-entailment, and ANT denotes antonym non-entailment samples. The best performing scores are **bolded**.

| Model | Acc. | CC Acc. | IC Acc. |
|---|---|---|---|
| BART | 62.48% | 61.87% | **63.10%** |
| BART-IEKG | **64.14%** | **65.24%** | **63.10%** |

Table 9: A comparison of the effect of IEKG injection on the FigurativeNarrativeBenchmark test dataset. Each sample has been split into a corresponding correct (CC) and incorrect sample (IC). The best performing scores are **bolded**.

semantic-wise similarity, but less so with token-wise similarity. Thus, similarity plays little role in the generated definition quality, but similar-meaning idioms are slightly more likely to receive better performance. This slight correlation is also seen on the scatter plots between the generation BERTScores and semantic-wise similarity, as shown in Figure 3. Note that correlation here denotes the Pearson correlation coefficient.

Finally, note that in many cases, we observed that low scores were not necessarily an indicator of low generalizability. Given the automatic nature of the ground truth idiom definitions, in many instances, definitions were either too short, had a much different length compared to the generated definition, or contained synonyms with the generated definition but exhibited little to no token overlap. Thus, many samples with a good definition may have ended up with poor BERTScores. As an example, the idiom "*in the raw*", with a ground truth definition of "*in its true state*", had a generated definition of "*to be unsophisticated or unrefined*". Despite this definition, this sample's BERTScore (precision) was $-0.2669$. This was just one of many samples we examined to have a good definition, but poor overall score. Therefore, we cannot claim that low similarity (either token-wise or semantic-wise) necessarily proves low generalization performance.

## H IE Comprehension Tasks with IEKG-trained KMs

This section demonstrates that IEKG-trained KMs can also improve the performances for downstream IE comprehension tasks. We show how our KMs enhances PTLM's IE-related knowledge comprehension ability via two concrete tasks related to IEs and figurative language, i.e., the *IE comprehension test* and the *figurative language inference*

### H.1 IE Comprehension Test

In English language learners, the ability to understand and use idioms is tested using idiom comprehension tests, which we adapt here to test PTLMs' idiom-related reasoning using IEKG.

#### H.1.1 Task and Data

Here, the comprehension ability is tested by asking a learner to select the correct continuation of the first half of a sentence containing an idiom. As shown in Table 13, given the first half of a sentence with an idiom, three possible continuations are presented, and the learner is asked to select the option that coheres with the meaning of the idiom. Note that the wrong options misunderstand the idiom's meaning in the context sentence, not necessarily confusing the literal and figurative interpretations. The idiom comprehension test dataset was collected from an idiom study resource by (Errey, 2018) consisting of 587 unique idiom instances, of which 131 (22.3%) are covered by IEKG. Hence, even with the additional knowledge from IEKG, models need to generalize to out-of-coverage idioms to perform well in this task.

#### H.1.2 Models

We consider a PTLM and two trained KMs as models for performance comparison.

**LM-BART** the pretrained BART-large language model in its off-the-shelf mode; we consider this

| Base Model | Overall Acc. | E Acc. | NE Acc. | ANT Acc. |
|---|---|---|---|---|
| IMPLI IEKG Samples | **72.60**% | **93.06**% | **61.90**% | **51.50**% |
| IMPLI Unseen Samples | 69.57% | 90.52% | 56.07% | 48.59% |
| FNB IEKG Samples | 62.25% | **65.41**% | 63.08% | N/A |
| FNB Unseen Samples | **63.98**% | 65.00% | **63.13**% | N/A |

Table 10: A comparison of BART-base performance on samples with IEs covered and uncovered by IEKG. Note that E denotes entailment, NE denotes non-entailment, and ANT denotes antonym non-entailment. For the FNB dataset, E denotes the correct continuation samples, and NE denotes the incorrect continuation samples. The best performing scores are **bolded**.

| Large Model | Overall Acc. | E Acc. | NE Acc. | ANT Acc. |
|---|---|---|---|---|
| IMPLI IEKG Samples | **84.65**% | **96.21**% | **69.40**% | **78.54**% |
| IMPLI Unseen Samples | 82.39% | 95.73% | 66.36% | 74.65% |
| FNB IEKG Samples | 77.77% | **78.49**% | 77.05% | N/A |
| FNB Unseen Samples | **77.81**% | 78.44% | **77.19**% | N/A |

Table 11: A comparison of BART-large performance on samples with IEs covered and uncovered by IEKG. Note that E denotes entailment, NE denotes non-entailment, and ANT denotes antonym non-entailment. For the FNB dataset, E denotes the correct continuation samples, and NE denotes the incorrect continuation samples. The best performing scores are **bolded**.

| Similarity | Precision | Recall | F1 |
|---|---|---|---|
| TW | 0.0243 | 0.0487 | **0.0380** |
| SS | **0.1324** | **0.1860** | **0.0380** |

Table 12: A comparison of BERTSCore correlation with both token-wise (TW) and semantic-wise similarity (SS). The best performing scores are **bolded**.

| Context Sentence | Options |
|---|---|
| Angelo saw it as a vote of confidence when his boss | gave him a bonus (✓) |
|  | told him to work harder |
|  | dismissed him |
| Tell your kids to steer clear of that dog. It | bites people (✓) |
|  | loves children |
|  | wags its tail |

Table 13: Examples of context sentences and their continuation options from the Idiom comprehension test dataset (correct answers marked by ✓).

model to compare the ability of a language model to that of the knowledge models.

**KM-ATOMIC** is the BART-large model trained on the ATOMIC$_{20}^{20}$ KG with the officially released checkpoint after one epoch of training.

**KM-Full** is a pretrained BART-large model trained on *both* ATOMIC$_{20}^{20}$ and IEKG tuples with the tuple completion task for two epochs with the same hyper-parameter settings as the KM-ATOMIC model. Comparing KM-Full to KM-ATOMIC allows us to assess the utility of the additional IEKG tuples.

### H.1.3 Methods

Our experiments are performed to gain insights into linguistic comprehension related to idioms and their ability to generalize knowledge from IEKG to idioms that are not in IEKG. In this experiment, we perform the comprehension test in a *zero-shot* classification setting, where the LM or a trained KM directly works on the comprehension task without being fine-tuned on any IE comprehension data. Using the following methods, we adapt LM and KM differently to perform zero-shot classification.

**LM Method.** For LM-BART, we concatenate a given context sentence and each of its three potential continuations and compute the log-loss for each continuation. The continuation with the lowest log loss is taken as the classification result. This zero-shot method has been used in (Chakrabarty et al., 2022a) for other figurative language interpretation tasks.

**KM Method.** For both the KMs, we turn the tail prediction task into a classification task to choose the best continuation as follows. First, we observe that most continuations in the dataset pertain to the subject of the given context sentence. Based on this, we select a set of six subject-related relation types in the experiment, i.e., we select $\mathbf{R}$ = [xAttr, xEffect, xIntent, xNeed, xReact, xWant]. Next, given $\mathbf{R}$, a set of $k$ relation types, and a context sentence $S$ with each of its three options of continuations $O_i, i \in 1, 2, 3$, we use the KM to consider the likelihood (correctness) of $O_i$ to be the likelihood

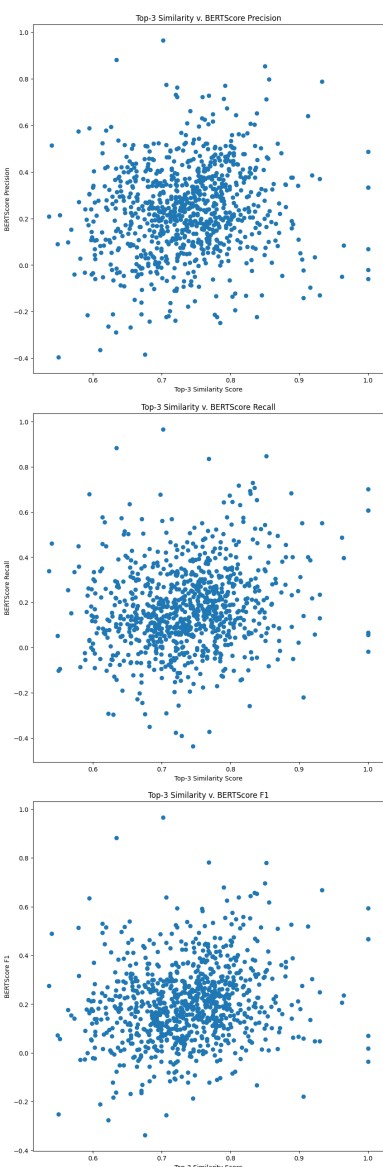

Figure 3: Scatter plots of generated definition BERTScores (precision, recall, F1) with idiom semantic-wise similarity.

of the knowledge tuple formed by $< S, r_j, O_i >$, such that $r_j \in \mathbf{R}$ and $j \in 1, ..., k$. Specifically, for each relation type $r_j$, we use the KM to compute three log-losses for the three tuples $< C, r_j, O_i >$ where $i \in 1, 2, 3$ and the $O_i$ with the smallest log-loss is taken as the classification result for the relation type $r_j$. Following this process, we produce a classification result for each of the $k$ relation types and then take a majority vote to produce the final classification result for the context sentence $S$. In other words, for each relation type $r_j$, we apply the *LM method* on the three continuation-converted tuples $< C, r_j, O_i >$ where $i \in 1, 2, 3$, and in the end, the classification result is the one chosen as

Table 14: Knowledge models' performance on the comprehension test in accuracy (%) for IEs. The results are broken down by IEs covered (Cov.) and uncovered (Uncov.) by IEKG. The best performances are **bold-faced**. (E1 and E2 refer to training in 1 and 2 epochs, respectively)

| Model | Overall | Cov. | Uncov. |
|---|---|---|---|
| LM-BART | 37.65 | 39.69 | 37.06 |
| KM-ATOMIC | 45.32 | 44.27 | 45.61 |
| KM-Full E1 | **50.60** | 50.38 | **50.65** |
| KM-Full E2 | **50.60** | **51.15** | 50.43 |

the correct continuation by the most relations.

### H.1.4 Results

As shown in Table 14, LM-BART cannot perform the IE comprehension test under the zero-shot setting, achieving an accuracy of 37.6%, which is barely above the random baseline accuracy of 33.3% (choosing one of three choices at random). LM's poor performance in the zero-shot setting is expected and aligns with prior results in similar classification settings (Hwang et al., 2021; Chakrabarty et al., 2022a), showing that a PTLM lacks the ability for commonsense and figurative language reasoning without further fine-tuning.

On the other hand, KMs are more competent in this task. Despite learning from a KG with a relatively smaller IE coverage and idiomatic annotations, KM-ATOMIC can achieve a 45.32% accuracy, which is a 7.67% accuracy gain over LM-BART. After training with the knowledge tuples from both ATOMIC and IEKG for only one epoch, KM-Full attains an accuracy of 50.6%, with an additional gain of 5.28% over KM-ATOMIC and an overall gain of 12.95% over the LM-BART. Also, we note that KM-Full's accuracy for IEs that IEKG does not cover is slightly above 50%, which is similar to the accuracy for the covered IEs. This result shows that KM-Full's comprehension ability is not restricted only to IEs that are seen from IEKG. Though the KM-full's accuracy is relatively low in number, we stress that this performance is achieved without any supervised training, i.e., KM-Full passes the comprehension test for half of the IEs in the test data in a zero-shot fashion. We expect the KMs to achieve much higher performance with a larger dataset for the comprehension test and supervised fine-tuning. Taken together, these results demonstrate (1) KM-Full's ability to carry out the IE comprehension test for certain IEs without

being fine-tuned to the task and (2) the KM's ability to generalize its comprehension to IEs unseen during KM training.

One limitation of the current zero-shot method is that every relation type is weighted equally; however, some relation types could be more contextually appropriate than the other relation types and thus should be considered with a higher weight. Observing the errors from KM-Full, we also found that the subject-focused relation types are sometimes inappropriate since the continuation is regarding the actions of the object of the sentence. We leave the exploration of potentially more capable zero-shot methods to future research as it is not the main focus of this study.

## H.2 Figurative Language Inference

### H.2.1 Task and Data

We use the FLUTE dataset (Chakrabarty et al., 2022b), with 7,534 NLI instances covering four types of figures of speech, i.e., idiom, metaphor, sarcasm, and simile. In particular, the data contains 479 idioms with a ∼58% (278) overlap with IEKG. Hence, to perform well, the model needs to generalize to idioms not covered by IEKG. Each instance consists of a premise sentence and a hypothesis sentence that includes a figure of speech; the task is to predict if the premise entails or contradicts the hypothesis. We reserve 500 random instances as the test set.

### H.2.2 Method and Models

To demonstrate the usefulness of the inferential knowledge afforded by IEKG, we fine-tune and compare three base version BART models to directly output '*Entailment*' or '*Contradiction*' to perform this task:

(1) **BART-Base**: We format the input instance as

   "*Premise:* [P]  *Hypothesis:* [H]"

where [P] and [H] are the premise and hypothesis sentence, and  is a separator token;

(2) **BART-ATOMIC**: We test the performance of the KM that is trained with the original ATOMIC$_{20}^{20}$. We take the ATOMIC$_{20}^{20}$ KG with the officially released checkpoint as the inference KM. Then, we use the KM to produce two inferences on each of the six relation types, including $\mathbf{R} = $ [xAttr, xEffect, xIntent, xNeed, xReact, xWant], for the hypothesis sentence in the FLUTE instance, resulting in twelve inferences in total. We then convert the inferences into a single sentence using the template shown in Table 5 connecting

inferences for the same relation type with the word 'and'; we format each instance as

"*Premise:* [P]  *Hypothesis:* [H]  *Inference:* [I]"

where [I] is the sentence inferred by the KM; Once we convert all input instances into the above format, we fine-tune a base version BART model using the input instances with the inferences and their corresponding output classes.

(3) **BART-IEKG**: In this method, we test the usefulness of the additional IE-related tuples in IEKG. We first convert a BART-large model into KM by training the model with the knowledge tuple completion task on tuples from *both* ATOMIC$_{20}^{20}$ and IEKG; the model is trained until convergence. Then, we use this new KM to replace the KM used in the BART-ATOMIC method to produce inference sentences and reformat the input sentences. Finally, we fine-tune a base version BART model with the inputs with new inference sentences.

All three models are trained for 25 epochs with a batch size of 32, a learning rate of 1e-5, an AdamW optimizer, and other hyperparameters in default.

### H.2.3 Results and Discussion.

Table 15 presents the results from the best-performing checkpoints. BART-IEKG outperforms BART-BASE by 6.20% in overall accuracy; specifically, there is an accuracy gain of 5.61% for idioms, 8.97% for metaphors, and 20.73% for similes. Because the official test set is not released by Chakrabarty et al. (2022b), we report that our results have comparable performance trends with theirs, e.g., models perform the best on sarcasm. Furthermore, we find that BART-IEKG performs well for both the "contradiction" (0.93 F1 score) and the "entailment" (0.90 F1 score) classes, gaining 6% and 7% in F1 respectively over to BART-ATOMIC. We identify that BART-IEKG correctly classifies 9% of test instances (9.8% of idiom instances) that are misclassified by BART-ATOMIC, while BART-ATOMIC only outperforms BART-IEKG on 2.8% of instances (4.5% of idioms instances). A qualitative analysis shows that BART-IEKG produces more accurate inferences when it outperforms BART-ATOMIC. For example, for the hypothesis on "*cross examine the witnesses*", BART-IEKG infers the narrator is *attentive and dutiful* and intended to *know the truth* while BART-ATOMIC infers the narrator is *assertive* and intended to *get their point across*. The improvement in similes is also significant. We attribute this to the

Table 15: Performances for NLI task with figurative speech in terms of overall accuracy (%) and accuracy by each figurative speech type. The best performances are **bold-faced**.

| Model | Idiom | Metaphor | Sarcasm | Simile | Overall |
|---|---|---|---|---|---|
| BART-BASE | 81.31 | 73.08 | 98.28 | 70.73 | 86.20 |
| BART-ATOMIC | 80.37 | 74.36 | 98.28 | 64.63 | 85.20 |
| BART-IEKG | **85.98** | **83.33** | **98.71** | **85.37** | **91.40** |

syntactic similarity between idioms and similes—many idioms are conventionalized similes that follow the same structure of using "like" or "as", e.g., *drink like a fish* and *drunk as a lord*; 48 IEs in IEKG follow this structure. In short, KMs trained with the additional IEKG tuples are more capable of comprehending and producing useful inferential knowledge on figurative language compared to when trained on $ATOMIC_{20}^{20}$. Additionally, the ability generalizes to not only unseen idioms but also to figures of speech beyond idioms.

Finally, we also note that our KM makes literal interpretations and sometimes misses the figure of speech, especially in longer sentences, where the KM infers over other events in sentences. We attribute this to the KM's training on short idiomatic events containing single IEs. Future research should explore other ways of combining KG and PTLMs to improve on natural (non-event) sentences.