# OpenReview forum: "IEKG: A Commonsense Knowledge Graph for Idiomatic Expressions"
_EMNLP/2023/Conference — EMNLP 2023 Main_

### Official Review · Reviewer_qKwD · 2023-08-04

**Soundness:** 4

**Excitement:**

3: Ambivalent: It has merits (e.g., it reports state-of-the-art results, the idea is nice), but there are key weaknesses (e.g., it describes incremental work), and it can significantly benefit from another round of revision. However, I won't object to accepting it if my co-reviewers champion it.

**Paper Topic And Main Contributions:**

This paper addresses the issue of Idiomatic expression (IE) comprehension by constructing an IEKG (Idiomatic Expression Knowledge Graph). The design of IEKG is extensively described in the article.

Using this dataset to enhance various Pre-trained Language Models (PLMs) shows significant improvements in performance on downstream tasks.

**Reasons To Accept:**

1. This paper addresses the issue of Idiomatic expression (IE) comprehension by constructing an IEKG (Idiomatic Expression Knowledge Graph). The design of IEKG is extensively described in the article.
2. Using this dataset to enhance various Pre-trained Language Models (PLMs) shows significant improvements in performance on downstream tasks.

**Reasons To Reject:**

It is batter to utilize a greater number of downstream tasks to evaluate the effectiveness of the dataset.

**Reproducibility:**

5: Could easily reproduce the results.

**Reviewer Confidence:**

3: Pretty sure, but there's a chance I missed something. Although I have a good feel for this area in general, I did not carefully check the paper's details, e.g., the math, experimental design, or novelty.

---

> ### Author Rebuttal · Authors · 2023-08-28
>
> ***Question**: It is batter to utilize a greater number of downstream tasks to evaluate the effectiveness of the dataset.*
>
> Thank you for your review and suggestions. Our primary objective with the application experiments is to demonstrate the benefits of incorporating IEKG to enhance language models' comprehension abilities for idiomatic expressions. As a proof of concept, we applied our IEKG injection with BART to four different tasks (two in the main paper and two in the Appendix) specifically designed to assess models' IE comprehension abilities.
>
> The main reason we did not include more downstream tasks is the limited availability of publicly accessible benchmark datasets for idiom comprehension tasks. The datasets we used in this paper, namely the IMPLI dataset for NLI and the Chakrabarty et al. dataset for context continuation, are currently the largest and most well-defined datasets for figurative language comprehension. In Appendix H, we also described experiments conducted on two additional datasets: (1) an IE comprehension test where, given the first half of a sentence containing an idiom, models are asked to select the continuation that aligns with the idiom's meaning from three possible options in a zero-shot fashion, and (2) the FLUTE NLI dataset, which encompasses a broader range of figurative languages, such as metaphor, simile, sarcasm, and idiom. In these experiments, we observed similar results to those presented in the main paper: models equipped with IEKG knowledge injection exhibit enhanced figurative language comprehension abilities. Due to the limited availability or smaller size of these datasets, we excluded them from the main paper to comply with the page limit.
>
> We did not include generic NLU datasets like SNLI, MNLI, or SST because these datasets are not specifically designed to assess a model's ability to comprehend idiomatic languages. For instance, SNLI contains a small number of sentences featuring idiomatic expressions. From these sentences, we would need to manually annotate (1) whether the idiomatic expressions are used in their figurative sense instead of their literal sense, and (2) if understanding the idiomatic expressions is crucial for classifying the given instance. Unfortunately, we do not currently possess the resources to perform such manual filtering of these datasets, and creating datasets to test model comprehension abilities is beyond the scope of this paper. Nevertheless, we believe that our existing experiments and tasks effectively demonstrate the benefits of IEKG in enhancing IE comprehension.

---

### Official Review · Reviewer_vAg2 · 2023-08-05

**Soundness:** 3

**Excitement:**

3: Ambivalent: It has merits (e.g., it reports state-of-the-art results, the idea is nice), but there are key weaknesses (e.g., it describes incremental work), and it can significantly benefit from another round of revision. However, I won't object to accepting it if my co-reviewers champion it.

**Paper Topic And Main Contributions:**

This paper constructs a commonsense knowledge graph for idiomatic expressions (IEKG) based on ATOMIC KG.

**Reasons To Accept:**

This paper describes the process of knowledge graph constraction, including idiomatic event creation, relation type selection, and the assessment of the annotation. Experimental results on IEKG injection, natural language inference, and context continuation show that neural model with this knowledge graph perform better than the model with ATOMIC.

**Reasons To Reject:**

This paper only compared large models trained on proposed IEKG, and compared the models with/without IEKG on one dataset for natural language inference, and context continuation tasks . How about the performance of other models and on other datasets, such as SNLI and MNLI.

**Reproducibility:**

4: Could mostly reproduce the results, but there may be some variation because of sample variance or minor variations in their interpretation of the protocol or method.

**Reviewer Confidence:**

2: Willing to defend my evaluation, but it is fairly likely that I missed some details, didn't understand some central points, or can't be sure about the novelty of the work.

---

> ### Author Rebuttal · Authors · 2023-08-28
>
> ***Question**: This paper only compared large models trained on proposed IEKG, and compared the models with/without IEKG on one dataset for natural language inference, and context continuation tasks How about the performance of other models and on other datasets, such as SNLI and MNLI.*
>
> Thank you for your review and suggestions. In Section 4, we evaluated various pretrained language models capable of conditional generation, such as GPT-2, T5, and BART, for our experiments. BART demonstrated the best performance, which is consistent with the results from the original ATOMIC paper. Consequently, we chose BART as our backbone model for further experimentation. Our primary objective with the application experiments is to demonstrate the advantages of incorporating IEKG to enhance language models' comprehension abilities for idiomatic expressions. We believe that the benefits and usefulness of IEKG are evident even with only BART in our experiments.
>
> Regarding publicly available benchmark datasets for idiom comprehension tasks, we included two datasets and tasks in the main paper, and the reason more are not included is that there are very limited options. The datasets we utilized in this paper, namely the IMPLI dataset for NLI and the Chakrabarty et al. dataset for context continuation, are currently the largest and most well-defined datasets for figurative language comprehension. In Appendix H, we also described experiments conducted on two additional datasets: (1) an IE comprehension test where, given the first half of a sentence containing an idiom, models are asked to select the continuation that aligns with the idiom's meaning from three possible options in a zero-shot fashion, and (2) the FLUTE NLI dataset, which encompasses a broader range of figurative languages, such as metaphor, simile, sarcasm, and idiom. In these experiments, we observed similar results to those presented in the main paper: models equipped with IEKG knowledge injection exhibit enhanced figurative language comprehension abilities. Due to the limited availability or smaller size of these datasets, we excluded them from the main paper to comply with the page limit.
>
> As for the absence of generic NLU datasets like SNLI, MNLI, or SST in the paper, these datasets are not specifically designed to assess a model's ability to comprehend idiomatic languages. For instance, SNLI contains a limited number of sentences containing idiomatic expressions. From these sentences, we would need to manually annotate (1) whether the idiomatic expressions are used in their figurative sense instead of their literal sense and (2) if understanding the idiomatic expressions is crucial for classifying the given instance. Unfortunately, we do not currently possess the resources to perform such manual filtering of these datasets, and creating these NLU datasets is beyond the scope of this paper. Nevertheless, we believe that our existing experiments and tasks effectively demonstrate the benefits of IEKG in enhancing IE comprehension.

---

### Official Review · Reviewer_Ch1Z · 2023-08-07

**Soundness:** 5

**Excitement:**

4: Strong: This paper deepens the understanding of some phenomenon or lowers the barriers to an existing research direction.

**Paper Topic And Main Contributions:**

The paper introduces the (American-English) Idiomatic Expression Knowledge Graph (IEKG) which extends the existing knowledge graph  ATOMIC 20/20 (Hwang et al. 2021) by 1,229 idiomatic events that are derived from idiomatic expressions in MAGPIE (Haagsma et al. 2020), annotated with up to 11 ATOMIC relations types and up to 4 relation "tails" by 3 annotators, resulting in 56,315 instances of <event, relation, tail>. The idiomatic events are restricted to events that contain a human referent as subject and/or object, which also limits the relations types to 11 out of the original ATOMIC relations.

The dataset is evaluated for its linguistic diversity in terms of ROUGE scores and claim that gain in quality compensates for the loss in diversity.

Downstream evaluations by BART, GPT2, T5 models fine-tuned on IEKG in a knowledge tuple completion task, evaluated by 3 human annotators show that the best model (BART) learned to generalize even to unseen idiomatic expressions

The paper adapts a masking task as an injection method for "small frame" LLMs like BERT and BART such that they can be further fine-tuned for other tasks. Downstream evaluation of different injected/fine-tuned models for natural inference (that is operationalized as a binary classification task) on the IMPLI dataset (Stowe et al. 2022) produce SOTA results.

The limitation that the models are not explicitly tested on their handling of non-idiomatic readings of idiomatic expressions is addressed by the authors in the future work and limitation sections.





**Reasons To Accept:**

* Introduction of IEKG, a useful resource for English NLP that captures non-compositional readings of (multi-word) idiomatic expressions which are not captured well in "small frame" LLMs like BERT and BART due to their non-compositional semantics
* A thorough description of the annotation process and the resulting dataset including quality control measures that evaluated the linguistic diversity of the annotated "tails" and assurance of non-toxicity.
* The usefulness of the resource is demonstrated by the downstream evaluation of three "small frame" LLMs that are fine-tuned on IEKG in a knowledge tuple completion task and by experiments on the performance of LLMs injected with IEKG knowledge and fine-tuned for an NLI classification task.
* Adaptation of a masking task as an injection method
* The paper is written in a very clear way and provides thorough discussions of all evaluation results and also the limitations of the dataset and the evaluations (so far not tested to distinguish between idiomatic and non-idiomatic readings).

**Reasons To Reject:**

* None

**Reproducibility:**

4: Could mostly reproduce the results, but there may be some variation because of sample variance or minor variations in their interpretation of the protocol or method.

**Reviewer Confidence:**

2: Willing to defend my evaluation, but it is fairly likely that I missed some details, didn't understand some central points, or can't be sure about the novelty of the work.

**Typos Grammar Style And Presentation Improvements:**

* Please state early in the paper that the resource is for (American) English (in addition to your discussion in the Ethics section that I appreciate).
* 043ff.: Could you please explain the example from Balahur et al. (2010) in more detail for non-native readers? What is the idiomatic reading of “stirred up a hornest’s nest”? Is it supposed to have a positive or a negative sentiment?

Typos:

* 098 “IEs ( 20%)”  => delete space after “(“
* Faulty reference: Moon et al., 1998"  => is single-authored Moon, 1998: "Rosamund Moon. 1998. Fixed Expressions and Idioms in English: A Corpus-Based Approach. Oxford => also in the bibliography.
* 163 “However ...” => There is something grammatically wrong with this sentence /incomplete?
* 242 “events 2.” => delete space before footnote index
* 370 and elsewhere “mean/std.” => mean/sd

Bibliography:

* There are some incomplete references ,e.g., Radford et al 2019, Rajani et al. 2014, Zhang et al. 2020b, Zhou et al. 2021 – also check for upper case in titles.

---

> ### Author Rebuttal · Authors · 2023-08-28
>
> ***Question**: Could you please explain the example from Balahur et al. (2010) in more detail for non-native readers? What is the idiomatic reading of “stirred up a hornest’s nest”? Is it supposed to have a positive or a negative sentiment?*
>
> Thank you for your review and insightful suggestions! The idiom "Stir up a hornet's nest" is used to convey the idea of causing trouble or inciting a commotion. If the figurative meaning of this idiom is not accurately understood, the model may fail to see the negative sentiment of the sentence. In our revised version, we will ensure that readers are provided with clear definitions of the idioms used in our examples.
>
> Additionally, we will clarify at the outset that our idioms have been sourced from the British National Corpus (BNC), and, as such, predominantly reflect British usage.

---

### Meta-Review · Area_Chair_sbhA · 2023-09-18

**Recommendation:** 4

**Metareview:**

This paper describes the introduction of IEKG, a commonsense knowledge graph extending ATOMIC for the figurative meaning of idiomatic expressions. After describing the annotation process, the authors present some experiments on the benefits on training LMs on IEKG with a knowledge tuple completion task, showing that all LMs fine-tuned with IEKG manage to outperform a BART Comet baseline. Additionally, they present an extrinsic evaluation of the resource, showing that IEKG injection leads to better performances both in a natural language inference and in a context continuation task.

All the reviewers agree on the usefulness of the proposed resource and appreciate the careful description of the annotation process and of the steps taken for quality control. The experimental results seem solid, with the incorporation of IEKG leading to consistent improvements in tuple completion and in the two downstream tasks.
A possible weakness that has been mentioned (R2 and R3) is that the authors could have evaluated their model on a larger number of tasks/datasets (although I must say that I am already positively impressed by the experiments included in the current version of the paper, which also includes additional model evaluation in the Appendix).

---

### Decision · Program_Chairs · 2023-10-07

**Decision:**

Accept-Main

**Comment:**

This paper describes the introduction of IEKG, a commonsense knowledge graph extending ATOMIC for the figurative meaning of idiomatic expressions. After describing the annotation process, the authors present some experiments on the benefits on training LMs on IEKG with a knowledge tuple completion task, showing that all LMs fine-tuned with IEKG manage to outperform a BART Comet baseline. Additionally, they present an extrinsic evaluation of the resource, showing that IEKG injection leads to better performances both in a natural language inference and in a context continuation task.

All the reviewers agree on the usefulness of the proposed resource and appreciate the careful description of the annotation process and of the steps taken for quality control. The experimental results seem solid, with the incorporation of IEKG leading to consistent improvements in tuple completion and in the two downstream tasks.
A possible weakness that has been mentioned (R2 and R3) is that the authors could have evaluated their model on a larger number of tasks/datasets (although I must say that I am already positively impressed by the experiments included in the current version of the paper, which also includes additional model evaluation in the Appendix).